

# Repetition velocity as a measure of loading intensity in the free weight and Smith machine Bulgarian split squat

Kaifang Liao[1,2,*], Chao Bian[3,*], Zhili Chen[2], Zhihang Yuan[2], Chris Bishop[4], Mengyuan Han[2], Yongming Li[2,5] and Yong Zheng[6]

[1] School of Sports Health, Guangdong Vocational Institute of Sport, Guangzhou, Guangdong, China
[2] School of Athletic Performance, Shanghai University of Sport, Shanghai, China
[3] Human Physiology and Sports Physiotherapy Research Group, Faculty of Physical Education and Physiotherapy, Vrije Universiteit Brussel, Brussels, Belgium
[4] School of Science and Technology, London Sport Institute, Middlesex University, London, United Kingdom
[5] China Institute of Sport Science, Beijing, China
[6] Physical Education Institute, China West Normal University, Nanchong, Sichuan, China
[*] These authors contributed equally to this work.

Corresponding authors
Yongming Li, liyongming@sus.edu.cn
Yong Zheng,
zhengyong7867@163.com,
zhengyong668@cwnu.edu.cn

## ABSTRACT

This study investigated the grouped and individualized load-velocity profile (GLVP *vs.* ILVP) in Bulgarian split squat using Smith machine and free weight. Seventy five recreational male lifters completed two incremental loading tests of Bulgarian split squat. Mean velocity was measured by a linear-position transducer (GymAware). Linear regression equation was applied to construct the GLVP and ILVP. The agreement of predicted %1RM and measured %1RM was assessed by a combination of intraclass correlation coefficient (ICC), coefficient of variation (CV), standard error of measurement (SEM) and Bland-Altman analysis. Acceptable validity was defined as ICC > 0.75, CV ≤ 10% and $p ≥ 0.05$ (a paired Wilcoxon signed-rank test). A very high level of inverse load-velocity relationships were demonstrated in Bulgarian split squat ($r = -0.92$) with free weights and a Smith machine. ILVP (ICC ≥ 0.98, CV ≤ 8.73%, $p ≥ 0.56$) was valid enough to predict the %1RM, but GLVP of both limbs revealed large CVs in free weights (CV: 15.4%,15.63%) and a Smith machine (CV: 11.24%, 12.25%). Cross-validation between the actual %1RM and predicted %1RM using free weights and a Smith machine ILVP was not acceptable ($p ≤ 0.03$, CV ≥ 14.07%). A very high level of inverse relationship were observed between %1RM and MV in Bulgarian split squat using free weights and a Smith machine, indicating individualized load velocity properties, and the ILVP showed high between-devices variability in both scenarios. Using velocity as a measure of loading intensity in Bulgarian split squat needs to consider the individualized load velocity properties, and difference between free weights and a Smith machine.

## INTRODUCTION

Appropriate exercise selection is one of the most important variants to the effect of strength training (*Kraemer & Ratamess, 2004*). Exercises can typically be categorized into bilateral or unilateral. Bilateral exercises (*e.g.*, squat, deadlift, bench press, *etc.*) are performed by both limbs working simultaneously, which have been largely considered as the core exercises to include in strength training regimes (*Gregory Haff & Travis Triplett, 2017*). In contrast, unilateral exercises (*e.g.*, Bulgarian split squat, split squat, *etc.*) are exercises which prioritize one limb over another, and are often used for the reduction of asymmetries (*Bettariga et al., 2022*) and injury prevention in strength training (*Liao et al., 2021*). However, some key movement patterns in daily life and sport-specific skills require unilateral movement competency (*e.g.*, change of direction, kicking, sprinting, walking, *etc.*) (*Liao et al., 2021*). Therefore, it could be argued that unilateral exercises may offer a closer representation to some of the movement patterns seen in sport, which in turn, may help to maximize the transfer of training effect (*Brearley & Bishop, 2019*).

The Bulgarian split squat is a unilaterally-biased exercise with a narrow medial-lateral base and reduced lumbar load in comparison with bilateral exercises such as the back squat. Recent cross-sectional evidence has shown that the Bulgarian split squat produces greater muscle activation in the gluteus maximus and hamstring muscles, than the back squat (*McCurdy, Walker & Yuen, 2018*), and greater vastus lateralis activation than the hip thrust, when using the same relative loads (*McCurdy et al., 2021*). In addition, chronic studies have found that the Bulgarian split squat has similar effects as the back squat for enhancing lower limb strength, but favorable adaptations for enhancing change of direction speed and unilateral power performance (*Fisher & Wallin, 2014*; *Gonzalo-Skok et al., 2017*; *Stern et al., 2020*; *Speirs et al., 2016*). Thus, this body of evidence indicates that the Bulgarian split squat is crucial for providing positive adaptation for physical characteristics in sport.

The load lifted is a key variant for manipulating the desired adaptation in strength training. Typically, the percentage of one repetition maximum (%1RM) has been the most widely used parameter to prescribe the load in strength training (*Gregory Haff & Travis Triplett, 2017*; *García-Ramos et al., 2019*). Despite its efficacy, the traditional %1RM assessment method suffers from a number of limitations. First, 1RM testing procedures are very time-consuming, which is likely impractical when working with large squads of athletes (*González-Badillo & Sánchez-Medina, 2010*). In addition, the day-to-day variability of maximal strength may not match the originally tested 1RM value, due to variations in sleep, nutrition and psychological factors (*Greig et al., 2020*). With this in mind, an alternative method is to use repetition velocity as a measure of loading intensity in real-time based on the strong relationship between %1RM and repetition velocity, also named velocity based training (VBT) (*Weakley et al., 2020*). A large number of studies have confirmed that there is a very high level of agreement between repetition velocity and %1RM in some primary bilateral exercises using free weights and a Smith machine, such as the squat ($R^2 = 0.95$) (*Sánchez-Medina et al., 2017*), bench press ($R^2 = 0.97$)

(*Sánchez-Medina et al., 2014*), and press up ($R^2 = 0.99$) (*Balsalobre-Fernandez, Garcia-Ramos & Jimenez-Reyes, 2018*). Theoretically, the load velocity profile (LVP) derived from its regression equation can also be used to measure the %1RM accurately. It should be noted that LVP is exercise-specific for biomechanical, joint kinematic, and muscle architecture differences (*Fahs, Blumkaitis & Rossow, 2019*). Therefore, using velocity to prescribe load one must first construct the LVP of the corresponding exercises.

There are two models for constructing LVP in exercises: one is based on a large sample dataset to establish a grouped LVP (GLVP), and the other is based on individual data to establish an individualized LVP (ILVP). Early studies tended to adopt the former approach, assuming that each exercise has a specific GLVP that can be universally applied to all individuals (*González-Badillo & Sánchez-Medina, 2010*; *Pareja-Blanco, Walker & Häkkinen, 2020*). However, recent research has indicated the individuality of LVP in exercises, suggesting that the use of a GLVP may lead to the loss of precise advantages in load quantification using VBT (*Balsalobre-Fernandez, Garcia-Ramos & Jimenez-Reyes, 2018*; *Luis Pestana-Melero et al., 2018*; *Thompson et al., 2021*). However, there is still a lack of evidence comparing the accuracy of predicting %1RM between GLVP and ILVP in Bulgarian split squat.

The applied devices (*i.e.,* free weight or Smith machine) and technique between the eccentric and concentric phases (*i.e.,* with or without a pause) are two other important factors in the VBT application. Most studies on exercises' LVP have been conducted in a Smith machine which only allows movement in a fixed trajectory and often with a pause during the selected exercise (*González-Badillo & Sánchez-Medina, 2010*; *Sánchez-Medina et al., 2017*), which eliminates the stretch-shortening cycle (SSC). The relevance here is that although this aids with the instability of performing free weight movements, athletes do not move purely in a single plane of motion; thus, such methods of training and test measures reduce the ecological validity for athletes. To the best of our knowledge, no study has cross-validated the same exercises' LVP of free weights and a Smith machine with a non-pause technique.

Therefore, the primary purpose of this study was to investigate the load-velocity relationship of the non-pause Bulgarian split squat using a Smith machine and free weights, comparing grouped *versus* individualized LVPs (GLVP *vs* ILVP). GLVP and ILVP was obtained from a group subjects and individual subject respectively. A secondary aim was to cross-validate ILVP of a Smith machine *vs* free weights during the Bulgarian split squat. We hypothesized that a very high level of inverse relationship would be demonstrated in both free weights and a smith machine Bulgarian split squat, the ILVP would provide an accurate estimation of the velocity related with the %1RM, but not GLVP, and the ILVP of a Smith machine Bulgarian split squat cannot be used to predict the %1RM in free weights.

## MATERIALS & METHODS

### Experimental approach to the problem

This cross-sectional investigation was used to determine the LVP of the Bulgarian split squat with free weights and a Smith machine. Subjects were required to randomly perform

the barbell Bulgarian split squat with incremental loads up to their 1RM to establish both ILVP and GLVP. A total of four visits were required including two familiarization sessions and two incremental load tests, separated by 72-hours. A linear position transducer was adopted to record the repetition mean velocity (MV). Each subject conducted all tests at similar time of the day in an attempt to prevent results from being impacted by potential circadian effects.

## Subjects

Seventy-five recreational male lifters (age 22.5 $\pm$ 2.7 yrs, height 175.6 $\pm$ 7.5 cm, body mass 72.1 $\pm$ 11.1 kg, 1RM of Bulgarian split squat with free weights 90.36 $\pm$ 15.66 kg) were recruited by interview. Inclusion criteria were at least two years of resistance training experience, and accustomed to performing the Bulgarian split squat using a proper technique. Before each testing session, subjects were required to refrain from alcohol, medicine, any strenuous exercises within 48 h. and sign the informed consent after a detailed explanation about the aims. This study was approved by the Ethics Review Committee of Shanghai University of Sport (No.: 102772021RT088).

## Procedures

The subjects were required to perform basic anthropometric measures and familiarization with the exercise protocol before the incremental tests. A power rack and a Smith machine without counterweights (Lipper, Nantong, China) were used to perform the Bulgarian split squat in accordance with previous research (*Helme, Emmonds & Low, 2022*). Briefly, the lead foot was positioned in a straight line with the rear foot elevated to a height of 40 cm on soft roller of a Bulgarian split squat support rack for all subjects. The distance of the support rack from the lead leg was equal to the distance from the pad to the lead heel when the subjects sit on the support rack with straight leg and heel on the floor. The heel of the front foot remained in contact with the ground at all times, and the knee of the rear leg descended below the height of the lead leg's knee, to reach a depth where the thigh of the lead leg was parallel with the ground during the bottom of the movement, and the patella of the lead leg directly above the toe line. The barbell rested across the top of the shoulders, as per the position adopted for a high-bar back squat technique (*Bishop & Turner, 2017*). The concentric phase was executed immediately following a controlled eccentric phase as fast and explosively as possible to maximize utilization of the SSC. The free weights and a smith machine were randomly chosen for the Bulgarian split squat, then left and right leg were randomly executed,

## Incremental load testing

After jogging at a speed of 5-8 km/h for three minutes, the subjects performed a series of dynamic stretches on the main muscles involved in the Bulgarian split squat for five minutes, then performed 2 $\times$ 5 barbell Bulgarian split squat repetitions without additional load (*i.e.,* 20 kg). After resting for three minutes, the incremental load test was performed (Supplementary S1). The initial load was barbell weight (20 kg for free weight and 30 kg for the Smith machine), and progressively increased by 20 kg until the repetition MV < 0.7 m/s with three repetitions and a three minute rest interval. Following this, increases
of 10 kg were applied when MV ranged from 0.7–0.5 m/s with two repetitions performed and a three minute rest interval. Finally, increases of 2-5 kg were applied when MV < 0.5 m/s was evident and until 1RM was achieved with a single repetition and a five minute recovery. The repetition MV was provided in real time to encourage the subjects to exert their maximum effort. Only the fastest MV at each load was selected for further analysis.

A linear position transducer (GymAware Power Tool Version 6.1; Canberra, Australia) was adopted to collect MV of barbell for each repetition. The retractable cables of the device was attached to the right side of barbell 65 cm away from the center, and the device was placed perpendicular to the movement track of the barbell according to the manufacturer's instructions. The device has been previously shown to be reliable and valid in measuring barbell velocity (ICC > 0.95, CV < 5%) in our laboratory. The data was connected *via* Bluetooth to a handheld tablet computer (iPad; Apple, Inc., Cupertino, CA, USA), displayed on a device-specific application (GymAware version 21.1). The sampling frequency of the device was >50 Hz.

## Statistical analyses

Data were described by mean values and SDs, and statistical significance was set at $p \leq 0.05$. The box scatter plot was used to identify the outliers, and the Shapiro–Wilk test was used for normality using a statistical software package (IBM SPSS Statistics 26; Chicago, IL, USA). A paired Wilcoxon signed-rank test were used to compare the 1RM difference between the Smith machine and free weights Bulgarian split squat. Pearson's product-moment coefficient ($r$) was used to examine the magnitude of the correlation between the repetition MV and %1RM of the whole sample. The evaluation criteria were 0.0–0.1 as trivial, 0.1–0.3 as low, 0.3–0.5 as moderate, 0.5–0.7 as high, 0.7–0.9 as very high, 0.9–1.0 as practically perfect (*Luis Pestana-Melero et al., 2018*). The scatterplots of %1RM ($y$) and MV ($x$) were created in Microsoft Excel, individualized linear regression equations were established for the left and right limbs separately, utilizing multiple loads and repetition velocities for each individual. However, grouped linear regression equations, incorporating loads and repetition velocities from all participants, were created to encompass both the left and right limbs, and the coefficient of determination ($R^2$) was calculated to explain variance of the correlation coefficients. $R^2$ was calculated as the ration of the regression sum of squares to the total sum of squares. The actual %1RM value obtained from the tests was used as the gold standard, and the corresponding velocity value was substituted into both individualized and grouped regression equation to obtain the estimated %1RM value. The difference between the measured and predicted %1RM value was tested using a paired Wilcoxon signed-rank test due to non-normally distributed data. The intra-class correlation coefficient (ICC) (based on single measure, two-way mixed-effect model), with the average difference of the predicted and real %1RM and its 95% CI by bland-Altman were used to assess the systematic bias, and the LoA (limited of agreements) was used to provide a range within the differences between the predicted and real %1RM with 95% CI. Random error was used to assess the precision of measurements due to random factor. ICC values were evaluated as excellent (ICC > 0.9), good (0.75–0.9), moderate (0.5–0.74) or poor ($\leq 0.5$) correlation (*Koo & Li, 2016*). The standard error of measurement (SEM =

SDd $\times \sqrt{1-\text{ICC}}$) and coefficient of variation (CV) were calculated to determine absolute reliability, with CV values $< 5\%$ as good, 5–10% as moderate, $>10\%$ as poor (*Banyard, Nosaka & Haff, 2017*). GLVP was used to calculate the %1RM for each leg separately, the predicted values and true values for both legs were then calculated individually to determine the CV. The ILVP obtained from the Smith machine was applied to free weight in Bulgarian split squat, while the ILVP obtained from free weight exercises was applied to the Smith machine. The consistency between predicted %1RM and actual %1RM was compared to assess the validity of cross-utilizing ILVP between these two equipment types. The acceptable validity for LVPs were considered as $p > 0.05$ (a paired Wilcoxon signed-rank test), ICC $\geq 0.9$ and CV $\leq 10\%$ (*Koo & Li, 2016*; *Banyard, Nosaka & Haff, 2017*).

## RESULTS

The subjects' mean 1RMs of free weights and a Smith machine Bulgarian squat were 90.37 $\pm$ 15.66 kg and 95.52 $\pm$ 15.75 kg respective when using left limb as lead leg, and 90.27 $\pm$ 15.32 kg and 97.81 $\pm$ 18.51 kg respectively when using right limb as lead leg. The 1RM of the Smith machine Bulgarian split squat was significant heavier than the free weights (left limb: MD = 3.41 kg, 95% CI [0.41–6.41] kg, $p = 0.001$; right limb: MD = 4.88 kg, 95% CI [1.79–7.96] kg, $p = 0.003$).

As shown in Fig. 1, the Pearson correlations for %1RM and MV ($r = -0.92$, $p = 0.001$) showed a significantly practically perfect inverse correlation. The grouped prediction linear regression equation for both a Smith machine and free weights showed a high and similar prediction capacity for %1RM accounting for 85% of the variance in %1RM. The grouped linear prediction equation could be obtained: the Smith machine %1RM = $-95.26 \times$ MV + 121.9, free weights %1RM = $-115.1 \times$ MV + 127.6.

As shown in Table 1, the ILVP for both a Smith machine and free weights showed a very high prediction capacity for %1RM accounting for mean 96% of the variance in %1RM, but GLVP accounting for only 85%. Paired Wilcox signed-rank test revealed a non-significant difference ($P > 0.39$), and ICC revealed an excellent correlation (ICC $\geq$ 0.75) between measured and predicted value by ILVP and GLVP both in Smith machine and free weight. The value of the CV for the ILVP was lower than 8.73%, but GLVP was higher than 11.2%, which indicated that the validity of ILVPs of Bulgarian split squat both in a Smith machine and free weights were acceptable, but the validity of GLVPs were unacceptable. The value of CV demonstrated that LVPs of a Smith machine of Bulgarian split squat had less variability than free weights.

As shown in Fig. 2, the value of the $R^2$ for the Bland-Altman plots of measured and predicted %1RM by ILVPs were lower than 0.01 for both a Smith machine and free weights. the systematic bias represented the average difference of measures between predicted and measured %1RM was close to 0, which indicated that the estimation has similar value as the real. Random errors ($\leq$5.83%1RM) for both a Smith machine and free weights Bulgarian split squat were low as compared to the measured %1RM, free weights was slightly higher than Smith machine.

Table 2 showed that the results between %1RM values (predicted and measured) for free weights (cross-validation: predicted %1RM from Smith machine individual equation) and

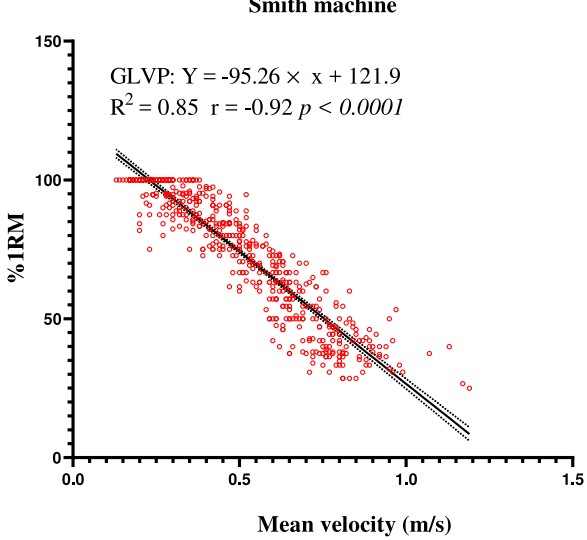

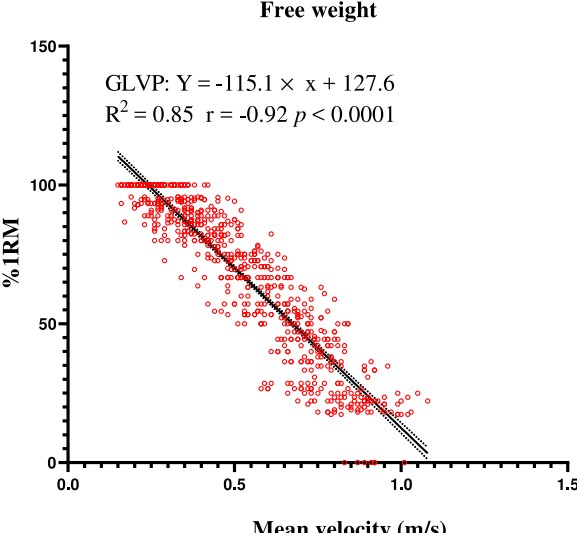

**Figure 1** **Relationship between the repetition mean velocity and relative load (%1RM) in Bulgarian split squat using a Smith machine and free weights.** The data included both limbs; GLVP, grouped load velocity profile.

a Smith machine (cross-validation: predicted %1RM from free weight individual equation) Bulgarian split squat were significantly different ($p < 0.03$), and the value of CV revealed a unacceptable high variability ($\geq 14.07\%$), which indicated that cross-validation between the actual %1RM and predicted %1RM using a Smith machine and free weights ILVP was not acceptable.

**Table 1   Validity between measured and predicted load in the Bulgarian split squat with Smith machine and free weight.**

| LVP | | Free weight | | Smith machine | |
|---|---|---|---|---|---|
| | | Left limb | Right limb | Left limb | Right limb |
| $R^2$ | ILVP | $0.96 \pm 0.03$ | $0.96 \pm 0.03$ | $0.96 \pm 0.04$ | $0.96 \pm 0.03$ |
| | GLVP | 0.85 | 0.86 | 0.84 | 0.87 |
| $P$ | ILVP | 0.91 | 0.71 | 0.56 | 0.71 |
| | GLVP | 0.39 | 0.68 | 0.8 | 0.6 |
| ICC/95%CI | ILVP | 0.98 (0.97–0.98) | 0.98 (0.97–0.98) | 0.98 (0.97–0.98) | 0.98 (0.98–0.99) |
| | GLVP | 0.92 (0.9–0.94) | 0.924 (0.91–0.94) | 0.914 (0.89–0.94) | 0.929 (0.91–0.94) |
| LoA (%1RM) | ILVP | $-11.27$–11.58 | $-11.03$–11.05 | $-9.44$–9.13 | $-8.31$–8.25 |
| | GLVP | $-20.49$–20.49 | $-20.15$–20.11 | $-17.35$–17.35 | $-15.94$–15.94 |
| SEM (%1RM) | ILVP | 4.12 | 3.98 | 3.35 | 2.98 |
| | GLVP | 7.39 | 7.26 | 6.26 | 5.75 |
| CV (%) | ILVP | 8.73 | 8.45 | 6.55 | 5.82 |
| | GLVP | 15.63 | 15.4 | 12.25 | 11.2 |

Notes.
LVP, load velocity profile; ILVP, individual load velocity profile; GLVP, group load velocity profile; $R^2$, coefficient of determination; ICC, intraclass correlation; LoA, limit of agreement; LoA, mean difference ± 1.96 random error; SEM, standard error of measurement; CV, coefficient of variability.

**Table 2   Validity of cross-validation on predicting %1RM from individual load velocity equation of Bulgarian split squat using a Smith machine and free weights.**

| | Free weight (cross-validation from Smith machine equation) | | Smith machine (cross-validation from free weight equation) | |
|---|---|---|---|---|
| | Left limb | Right limb | Left limb | Right limb |
| $P$ | 0.001 | 0.001 | 0.011 | 0.03 |
| ICC/95% CI | 0.922 (0.891–0.943) | 0.92 (0.894–0.939) | 0.906 (0.876–0.929) | 0.91 (0.883–0.932) |
| LoA (%1RM) | $-22.52$–16.27 | $-22.755$–17.83 | $-18.53$–22.88 | $-18.76$–21.09 |
| SEM (%1RM) | 7 | 7.32 | 7.47 | 7.19 |
| CV(%) | 14.5 | 15.29 | 14.77 | 14.07 |

Notes.
LVP, load velocity profile; ILVP, individual load velocity profile; ICC, intraclass correlation; LoA, limit of agreement; SEM, standard error of measurement; CV, coefficient of variability.

## DISCUSSION

The main purpose of this study was to determine and evaluate the agreement of the GLVP and ILVP, and cross-validated the ILVP of a Smith machine and free weights in the Bulgarian split squat. The primary findings of the current investigations were as follows: (1) %1RM and repetition velocity in the Bulgarian split squat using both the Smith machine and free weights revealed a very high level of inverse relationships, (2) the ILVP was valid enough to accurately predict the load, whereas the GLVP demonstrated unacceptable validity, and (3) ILVP of the Smith machine could not predict %1RM of the free weight condition accurately, and vice versa.

A very high level of inverse relationships ($r = -0.92$) were observed between %1RM and MV in the Bulgarian split squat exercise, using both the Smith machine and free weights. The lead foot has been identified as the primary contributing limb to total vertical

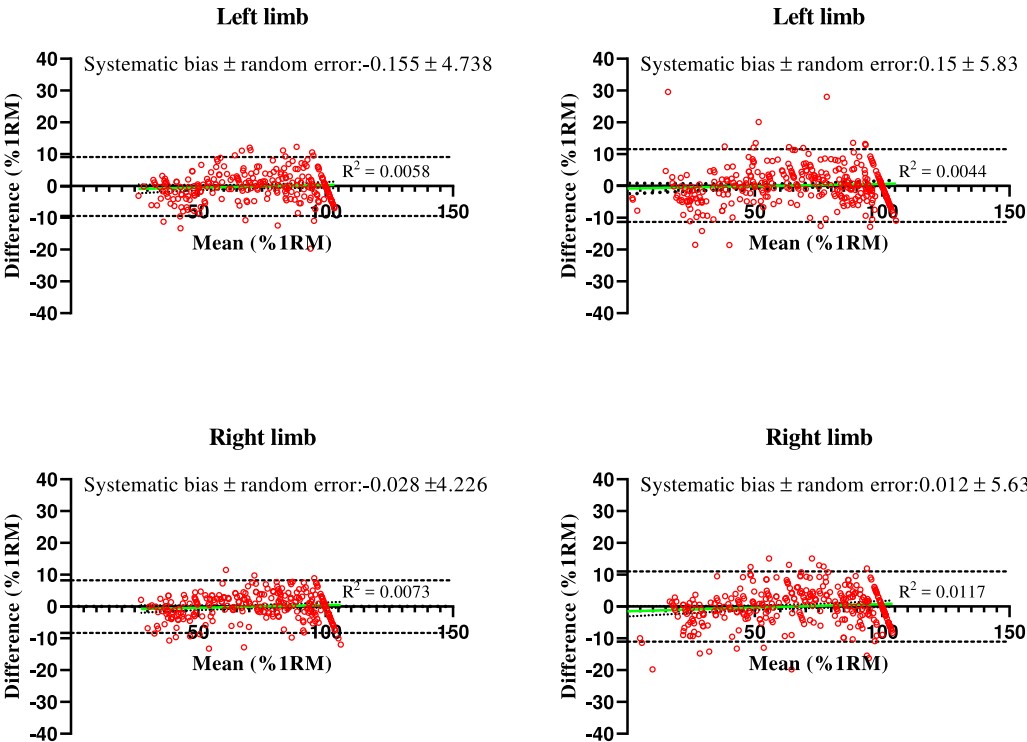

**Figure 2 Bland-Altman plots for the measured and predicted relative load (%1RM) by individual linear equation in Bulgarian split squat using Smith machine and free weight.** Difference = predicted %1RM—real %1RM; Random error is standard deviation divided by the square root of the number of measurements ($n = 2$).

force production (~85%) (*Helme, Emmonds & Low, 2022*), suggesting that the relationship between velocity and %1RM depends on the strength of the lead leg. Of note, our finding reported a weaker relationship ($R^2$: 0.85 *vs.* 0.95) than that obtained during the back squat by *Sánchez-Medina et al. (2017)*. A possible explanation for this might be due to the different execution techniques employed between studies, noting that *Sánchez-Medina et al. (2017)* imposed a pause during their back squat protocol. The relevance here is that a pause eliminates the SSC, which may have resulted in notable reductions in measurement error (*Pallarés et al., 2014*). However, the non-pause technique utilizing the SSC is likely to hold greater value to athletes given its prevalence in a wide variety of sporting movements (*e.g.*, sprinting, jumping, kicking, *etc.*). It is interesting to find that the *r* value of the Smith machine Bulgarian split squat was the same as the free weight condition, indicating that the different requirements of bar trajectory between the Smith machine and free weights may not affect correlation intensity of repetition velocity and %1RM.

The ILVP was valid enough to predict the %1RM, whereas GLVP showed an unacceptable validity in Bulgarian split squat using both free weights and a Smith machine. This finding
is consistent with that of *Thompson et al. (2021)* and *Banyard et al. (2018)* who both found that load-velocity characteristics were highly individualized in the free weight back squat, and that of *Balsalobre-Fernandez, Garcia-Ramos & Jimenez-Reyes (2018)* who observed large between-subjects variability across relative loads in the seated military press exercise performed in a Smith machine. The individuality of LVP might be attributed to the different limb biomechanics, muscle fiber type contribution, motor unit recruitment, and synergistic coordination (*Cormie, McGuigan & Newton, 2011*). For example, *Methenitis et al. (2016)* demonstrated that the % cross-sectional area of IIx fibers in vastus lateralis dictates a large part of the correlation between rate of force development and power performance in the leg press. Thus, the current findings draw our attention to the importance of considering the development of exercise's individualized LVP in VBT to induce optimal training effects.

The most interesting finding was that the cross-validation for ILVP between the actual %1RM and estimated %1RM using free weights and a Smith machine was unacceptable. Our finding was in line with that of *Loturco et al. (2017)* who also observed that ILVP of Smith machine had a better predictive accuracy than free weights in the bench press exercise. Unfortunately, *Loturco et al. (2017)* only compared the validity of ILVP between a Smith machine and free weights, but did not provide evidence on cross-validation between the ILVP using free weights and a Smith machine. To the best of our knowledge, this was the first study to cross-validate ILVP of the same exercise between a Smith machine and free weights. These results remind practitioners that strength training should not simply extrapolate the same exercise's ILVP obtained in a Smith machine to free weights. The fixed trajectory of the Smith machine undoubtedly restricts the movement path to the vertical direction, which produces different biomechanics from the same exercise performed using free weights (and without movement restrictions). For example, free-weight squats have significantly greater hip flexion and more torso forward lean than in a Smith machine when feet under the bar (*Gutierrez & Bahamonde, 2009*). In addition, with feet performed in a Smith machine, a greater focus on vertical force production can be applied, which may enhance the reliability of any recorded data. However, the stability of the trunk is often a limiting factor for producing the maximum force in free weight lower body exercises, especially in unilateral exercises demanding greater core stability (*Kibler, Press & Sciascia, 2006*), and is something that provides a mechanical advantage to doing unilaterally-biased exercises with free weights. Our findings also revealed that 1RM of the Bulgarian split squat in a Smith machine was heavier than when performed using free weights, which supported our suggestion that force production may be higher when the instability of an exercise is removed, which was consistent with *Cotterman, Darby & Skelly (2005)* and *Pérez-Castilla et al. (2020)*. Collectively, it seems that the differences in movement biomechanics and stability requirements, and the mis-match between 1RM and minimum velocity, contribute to the large variability in repetition velocities of the Smith machine and free weights at the same %1RM. Ultimately, this results in the ILVP's of the smith machine and free weights not being inter-changeable. Therefore, in practice, to ensure the accuracy of predicted %1RM of the Bulgarian split squat, practitioners should use the respective ILVP to measure the load in both the Smith machine and when performing with free weights.

This study has some limitations. Firstly, this study did not repeat the test of LVP, resulting in a lack of reliability evidence due to the participation of over 70 individuals and the inclusion of both the smith machine and free weights. Secondly, the subjects used in this study were recreational lifters with age $22.5 \pm 2.7$yrs, and whether the results can be generalized to professional athletes and other age group needs to be further verified. Future studies should seek to determine the reliability of the LVP in the Bulgarian split squat exercise, using more experienced athlete populations.

## Practical applications

VBT is currently a popular method to prescribe and monitor load in resistance training. The available exercises is a challenging for VBT application. Bulgarian split squat is a typical unilaterally-biased exercise which shown to be effective in developing athletic performance. The findings of the current study demonstrate that repetition velocity can be used as a measure of loading intensity in the Smith machine and free weights Bulgarian split squat. Therefore, the VBT practitioners are suggested to use Bulgarian split squat to improve their athletes' unilateral athletic qualities. In comparison of the predicted accuracy between GLVP and ILVP, the result supports the load velocity profile has individualized nature, thus, we recommend the practitioners should use an individualized approach to fit an athlete's LVP when implementing the non-pause Bulgarian split squat in VBT program. Despite this, this is the first study to cross-validate the accuracy of ILVP between Smith machine and free weights. The evidence suggest that practitioners should be cognizant of the difference in ILVP between free weights and a Smith machine, and use the respective ILVP to measure the velocity load in free weights and a Smith machine.

## CONCLUSION

A very high level of inverse relationships were observed between %1RM and MV in Bulgarian split squat using both free weights and a Smith machine. However, the respective larger and lower variability between predicted and measured %1RM of GLVP and ILVP indicated that load velocity property are individualized. The ILVP demonstrated high between-devices variability in free weights and a Smith machine Bulgarian split squat.

### Funding
This work was supported by the Guangdong Sport Bureau (No. GDSS2022N093). The funders had no role in study design, data collection and analysis, decision to publish, or preparation of the manuscript.

### Grant Disclosures
The following grant information was disclosed by the authors:
Guangdong Sport Bureau: GDSS2022N093.

### Competing Interests
The authors declare there are no competing interests.

## Author Contributions

- Kaifang Liao conceived and designed the experiments, performed the experiments, analyzed the data, prepared figures and/or tables, authored or reviewed drafts of the article, and approved the final draft.
- Chao Bian performed the experiments, prepared figures and/or tables, and approved the final draft.
- Zhili Chen performed the experiments, analyzed the data, prepared figures and/or tables, and approved the final draft.
- Zhihang Yuan performed the experiments, analyzed the data, prepared figures and/or tables, and approved the final draft.
- Chris Bishop conceived and designed the experiments, authored or reviewed drafts of the article, and approved the final draft.
- Mengyuan Han performed the experiments, prepared figures and/or tables, and approved the final draft.
- Yongming Li conceived and designed the experiments, authored or reviewed drafts of the article, and approved the final draft.
- Yong Zheng conceived and designed the experiments, prepared figures and/or tables, and approved the final draft.

## Human Ethics

The following information was supplied relating to ethical approvals (*i.e.*, approving body and any reference numbers):

The Ethics Review Committee of Shanghai University of Sport approved the study (No.: 102772021RT088).

## Data Availability

The raw measurements are available in the Supplemental Files.

## Supplemental Information

Supplemental information for this article can be found online at http://dx.doi.org/10.7717/peerj.15863#supplemental-information.

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
