# Peer review of "Repetition velocity as a measure of loading intensity in the free weight and Smith machine Bulgarian split squat"

_PeerJ, doi:10.7717/peerj.15863_

## Round 0.1 · original submission · Major Revisions

The authors submitted a well-thought-out and interesting research.

However, the manuscript should undergo a linguistic revision to improve its clarity, especially in the description of the methods and statistical analysis, which are sometimes unclear. Likewise, the clarity of the Figures, Tables, and dataset can be improved. Specifically, the authors should try to make the Figure, Tables, and dataset more self-explanatory. For example, the authors could better explain the title of the dataset columns and clarify its organization (e.g., are data on the same row coming from the same participant?).

Additionally, more information should be given on the methodology (participants, randomization procedures, and statistical analysis). In particular, the statistical paragraph needs some additional and clear explanations of the analyses performed and, when appropriate, their rationale. For example, the authors should justify how the presence of clusters in certain analyses (e.g., Fig 1 data from two limbs grouped) or type I error inflation were handled, or at least discuss these points.

Finally, the Practical applications section could be edited to focus more on the results of the paper (e.g., see the last three lines).

·

Basic reporting

Good on all points.

Experimental design

Good design adding new information with cross validation.
Specific comments:
Abstract-
Spell out 75 when starting the sentence.
CV and p 0.05
Line 36-the predicted
Line 88- Define individual vs group LVP. I am assuming that you mean taking both regression lines from the Smith Machine and Free Weight and combining to one but I am not sure.
Line 100- did not see MV spelled out prior
Subjects-Were these male and female?
Line 106-9 needs some grammatical revision


Procedures-
What occurred during the familiarization session?
How did you control the distance of the support pad from the lead foot?
Was the lead knee anterior/posterior translation in front or behind the lead toes controlled?
What specific part of the foot contacted the support Pad and what was the position of the ankle on the pad? What kind of support was used? A figure of the technique would help with the added description of the technique.
What was the average number of repetitions needed to reach the 1RM?
Line 146- Grammatical- you have several separate sentences running together.

Results
Line 186-189 Two sentences here to separate.

Validity of the findings

Statistical measures were sound. Had few comments for conclusions:
Discussion
Line 244 and reference 30- Bigger torso angle appears to suggest greater trunk flexion. Do you mean greater hip flexion and more torso forward lean? Trunk flexion would depend on the ability of the individual to stay neutral. I agree that primarily there is greater hip flexion reguired to center the weight over the base (feet) since you cannot do a back squat with a vertical trunk or you would have the center of gravity behind the feet and be a falling object. However, the stability provided by the smith machine allows large variability in technique. You can position your feet far in front of the bar and be very vertical or you have the feet under the bar and have more forward lean. These differences can also greatly affect the ankle dorsi-flexion angle that could be greater or less than a free weight squat. It would be important to discuss this.

Line 249- What do you mean by provides a mechanical advantage?
Line 280- Free weights being always more advantageous over the smith machine can be argued. In most cases and for athletes for the synergist activity etc that you mentioned but there are times when the smith machine can be an advantage. If you want to isolate the movement and produce max force in a more limited direction like you noted (vertical, etc). This is sometimes a variation or goal used even with highly trained or athletes. You can also more easily modify the technique since the load is more stable to target different joints or muscle groups similar to the foot position noted above. I suggest not using “always’ and may add some of this in the discussion.

Additional comments

Good and interesting data/work.

·

Basic reporting

Dear Authors and Editors:
I appreciate your work. VBT-related topics are currently heavily promoted in the media, so more reliable research is needed. However, the practical dimension of the recommendations based on the VBT potential is enormous, especially in the individualization of training loads in safe ranges of the speed of the weight movement.

I have some suggestions and recommendations that can help improve the quality of this work.

The structure and language of the work need linguistic and technical correction; there are many simple errors in the work:
such as a small letter in proper names when marking institutions) "3 human physiology and sports physiotherapy research group, faculty of Physical Education and Physiotherapy, Vrije Universiteit Brussel" ;
no commas
too many spaces.
no dots are needed, e.g. "vs"
plural instead of singular and vice versa
The entire manuscript needs corrections in terms of correctness of clarity. Here are some examples:

Line 47 - no comma "with"
Line 48 - suggests a change to the most important role
Line 50, 52 - The names of the exercises are listed in brackets (one plural and one singular) "bench press"
Line 51 - less verbose
Line 53 - no "the" before reduction
Line 56 - space
Line 65 - "favorable"
Line 72 - remove "to be"
Line 80 - less verbose "In turn, the " on the
Line 81 - more formally, e.g. to measure the % 1RM accurately
Line 83 - Replace the word "firstly" with "first"
Line 87 - change the wording "The majority of" to, e.g. "Most"
Line 92-93 - rewrite for clarity

There are many more of them in the whole work, and the reader should be fine with interpretation.

The literature contained in the introduction needs to exhaust the topic.
The introduction has no information about the standard warm-up procedure necessary to validate the day's instructions using VBT. It will complement the reader's awareness of the need to permanently plan such an element preceding the central part of the session.
Cross-validations and their implication in prediction systems are an exciting aspect of the analysis. Unfortunately, more information about this topic needs to be in the introduction, even from other sports.

Sometimes an additional reference would also strengthen the message
Line 56 - Literature reference needs to be included here.

On the Figure 1. Relationship between the repetition mean velocity and relative load (%1RM) in Bulgarian split squat using Smith machine and free weight. The data included both limbs.
The free weight %1RM to velocity graph will show 0% values - please explain this anomaly

Experimental design

There is no information about the rules that determine the group.
What does volunteer mean? The lack of correctly and clearly defined selection criteria for the research group makes it impossible to repeat and accurately draw conclusions. Lack of information on how the information about the experience of the respondents was collected? "Questionnaire? Interview? E.t.c"
Although there is information about the recreational level of the respondents at the end of the work, it should be found in the material and methods section.

Incremental load testing
Here you can specify the information: by pattern.
A) they ran on a treadmill at a speed of 5-8 km/h
B) they ran on a treadmill to a maximum heart rate of 55%
C) Their task was to run 500 m in 3 minutes.
The same applies to the stretching scheme.
You can add an attachment with a detailed flowchart of the preparation procedure. Without. This analysis cannot be repeated.

He proposes to place the course of the study on a time chart. It will be transparent.accurately,TheThistransparent accurately

Validity of the findings

Both in the conclusions and in the discussion, the topic of the specificity of the study group should be raised. Because of its absence, conclusions may be overestimated. And such high correlation results or the lack or error shown in work may differ from the population.
This should be improved.

Additional comments

The article contains material that can be a positive value for motor preparation trainers working on the basis of velocity-based training. However, it must be corrected, especially for shortcomings in the description of the research methodologies.

Good luck with your further research!
Michal Nowak, Ph.D
Jan Długosz University of Humanities and Natural Sciences in Częstochowa
Faculty of Health Sciences
educationsupport@icloud.com

Reviewer 3 ·

Basic reporting

I think the author clearly expressed the aim of the study and the background for it.
I suggest re-writing the first sentence in the introduction section. The sentence must be more precise and link better with the second one.
The authors provide clear tables and figures to explain the results.
I suggest inserting a figure or a scheme to better explain the incremental load testing.

Please remove "near perfect" or "perfect" from the results and discussion sections. "A very high level of agreement" or similar is more appropriate.

References are sufficient and recent.

Experimental design

No comment

Validity of the findings

No comment

Additional comments

The authors provided relevant information on the resistance training context.
I think the rationale and the hypothesis are well explained.
The conclusion is well stated, please also report the study's strengths and not only the limitation.

---

## Round 0.2 · Major Revisions

The authors have addressed most of the comments raised during the first round of revisions.

However, additional work is needed to improve the manuscript’s clarity from both a methodological and linguistic perspective, which is essential to meeting PeerJ publication standards.

Please, follow PeerJ guidelines (https://peerj.com/about/policies-and-procedures/ & https://peerj.com/about/author-instructions/), especially regarding the description of the methods and statistical analysis, and try to address the following comments:

Line 35. Should CV have the sign <? Additionally, which statistical test is the p-value referring to?

Line 38. Please, specify the reason why there are two CVs within the parentheses.

Line 38. Did the authors perform the statistical technique of cross-validation? If so, it is ok to leave it as it is in the abstract, but the statistical analysis performed should be specified and described in detail in the statistical analysis section.

Line 40-41. The conclusions of the abstract are not clear, please consider revising them and correcting the punctuation.

Line 106. Please, specify if the 1RM value comes from free weights or Smith machine.

Line 111-141. Please, clarify in detail how the randomization was performed (considering both limbs and exercise modalities).

Line 115. Please, specify what “straight line” is referred to.

Line 142-162. The statistical analysis and its rationale should be explained in more detail (please see PeerJ guidelines suggested above). In particular, the authors should address the following points:
a) Please, explain in more detail how the GLVP and ILVP were calculated and how the presence of clusters (e.g., data coming from the same participants [multiple loads and the two limbs]) was considered in the context of meeting the statistical assumptions of linear regressions (or at least discuss and justify it).
b) Please, explain in more detail how the presence of multiple testing and type I error inflation was taken into account.

Line 152. Please, review this sentence, assessing its accuracy from a statistical and linguistic perspective.

Line 158. Please, define how the systematic bias is assessed.

Line 162-163. Please, specify how the cross-validation was performed and correct the punctuation.

Line 163. Please, specify the statistical test the p-value is referring to.

Line 167. Should it be “respectively”?

Line 167. Please, add a space between the number and the measurement unit throughout the manuscript (e.g., 15.75 kg).

Line 175. Please, replace the * with the sign ×.

Line 191. Please, explain how the R2 is computed and consider if it is appropriate to use it in this context or if the correlation coefficient Pearson’s r should be preferred.

Line 194. Please, explain how the random errors were computed.

Line 219. Why are the findings weaker? Did the authors mean that the correlation/relationship was weaker?

Line 265-266. This sentence needs to be clarified, please consider revising it.

Figure 1. Is the figure referring to GLVP? If so, please specify it.

Figure 2. Please, specify how the difference and random error were computed.

Table 1. Please, specify the % levels (e.g., 95%) of the LoA.

Table 2. Please, specify what the term validity refers to (e.g., predicting %1RM).

·

Basic reporting

Abstract
1. Line 35: CV≤10% instead of CV>10
2. Last line needs capitalized
3. There needs to be a conclusion as the final sentences are stated results. In the concluding statements, how do we use these results?

Introduction:
1. Line 76: need to add “one” in front of “must first”…
2. Line 77: If this is referring to your study, it should go in the Methods but if this is referring to previous studies, state previous studies… and cite these. However, I suggest deleting since it does not appear to add to the justification. Explain why this is important to know if you are keeping this in the introduction. Line 89: also is part of the methods. Justify why this is important in the introduction but do not state your actual methods until the methods section.
3. Line 91-94: Sentence is wordy and too long. I suggest breaking it up to be more clear. In addition, you have a hypothesis about grouped vs individual differences in the results but there is no discussion about this in the introduction. Add justification in the introduction why you have this hypothesis.


Methods:
1. Line 99: barbell refers to free weight but also state that the smith machine was used. I assume that what you mean by randomly is that they performed the free-weight and smith machine in random order to develop the actual LVP for each. Correct?
2. Line 161: Capitalize the start of the sentence.

Discussion:
1. Line 251: replace when with “with feet…”
2. Line 267-9: wordy sentence. Consider to revise.
3. Line 279: add “thus, we recommend…”

References:
1. Reference 21 not complete in the reference list.

Experimental design

no comment

Validity of the findings

no comment

Additional comments

no comment

·

Basic reporting

The work has been corrected according to the instructions. I have no further comments for this section.

Experimental design

The work has been corrected according to the instructions. I have no further comments for this section.

Validity of the findings

The work has been corrected according to the instructions. I have no further comments for this section.

Additional comments

The work has been improved as suggested. After the corrections have been made, the work should be made public.

---

## Round 0.3 · Minor Revisions

I commend the authors for the effort put into revising the articles according to the given suggestions. The article has greatly improved. Please consider these minor comments:

Line 35. Should CV have the sign <? Additionally, which statistical test is the p-value referring to?
Response: Yes, The CV should have the sign “≤”, we have already corrected it. The P-value is referring to a paired Wilcox signed-rank test, we added this in the sentence. Thank you.
Editor response: In the manuscript there is written t-tests, which one is right? Please also verify the spelling of the signed-rank test, shouldn’t it be Wilcoxon?

Line 38. Please, specify the reason why there are two CVs within the parentheses.
Response: Because GLVP was used to calculate the %1RM for each leg separately, the predicted values and true values for both legs were then calculated individually to determine the coefficient of variation (CV).
Editor response: Please, specify this in the paper.

Line 163. Please, specify the statistical test the p-value is referring to.
Response: we specified it as Paired samples t-tests. Thank you.
Editor response: It is written Wilcox in the text.

Line 191. Please, explain how the R2 is computed and consider if it is appropriate to use it in this context or if the correlation coefficient Pearson’s r should be preferred.
Response: We use R2, because it could tell us the proportion of variance that the two variables have in common, and also R2 can identify whether there is a noticeable trend in the variation of differences with changing load, so we preferred R2 to r. Thank you.
Editor response: It makes sense, but please try to clarify and specify the variables used to compute R2, it should be easily understandable for the readers.

Line 194. Please, explain how the random errors were computed.
Response: the random error is Standard error of difference, which used to calculate the 95% CI of agreement limits allows for the estimate of the size of the possible sampling error. Thank you.
Editor response: please, uniform this with the description provided in the statistical analysis section. In the statistical analysis section, the random error is defined as the LoA. The authors can use both indicators to describe the random error, but it should be clear for the reader which one is it.

Line 219. Why are the findings weaker? Did the authors mean that the correlation/relationship was weaker?
Response: Because the R2 value of our finding were less than the finding by Sanchez-Medina. It actually mean that correlation/relationship was weaker.
Editor response: As I thought it was referring to the correlation/relationship, please rewrite the sentence specifying you are referring to that. Saying that the findings are weaker seems incomplete to me. You could say something like: “our founding reported a weaker relationship”.

Figure 2. Please, specify how the difference and random error were computed.
Response: thanks for this comments, we have already specified the difference and random error.
Editor response: In order to make the figure self-explanatory, the authors should specify what the random error expressed here is in the note of the figure.

Table 1. Please, specify the % levels (e.g., 95%) of the LoA.
Response: LoA is the limited area which calculated by the mean difference ± 1.96 random error (95%CI), which already included the %95 CI. Thank you.
Editor response: In order to make the figure self-explanatory and avoid unclarity, the authors should still specify the % used for the LoAs somewhere in the table.

---

## Round 0.4 · accepted · Accept

Dear authors,

Thank you for your efforts in revising the manuscript, which is now ready for publication.

Good luck with your future work.

Carlo